**Data Availability Statement:** All relevant data are within the paper and its Supporting Information files.

**Funding:** The author(s) received no specific funding for this work.

# The relationship between workplace bullying and family functioning: A systematic review

**Yang Jie**[1,2], **Daniella Mokhtar**[1]*, **Nurul-Azza Abdullah**[1]

**1** Centre for Research in Psychology & Human Well-Being, Faculty of Social Sciences and Humanities, The National University of Malaysia, Bangi, Malaysia, **2** Department of Psychology, School of Marxism, Nanchang Medical College, Nanchang, People's Republic of China

* daniellamokhtar@ukm.edu.my

## Abstract

While the occupational and health-related consequences of workplace bullying have received extensive research attention, the effects of workplace bullying on the family domain have been largely ignored. Based on the PRISMA framework, the Scopus, Web of Science, PsycINFO, and PubMed databases were searched up to May 12, 2024, for articles on associations between workplace bullying and family functioning. A total of 1347 articles were identified, of which 37 were found after review to meet the criteria for inclusion. All the included studies found a direct or indirect association between workplace bullying and family functioning. Most studies are grounded in the conservation of resources (COR) theory, spillover theory, crossover theory, and work-family interface model. Negative affect (emotions), work-family conflict (WFC), and burnout were considered essential mechanisms explaining the links between workplace bullying and family functioning, with personal resources (demands) as the main moderators. Most studies focus on the one-way impact of workplace bullying on family functioning, mainly using cross-sectional, non-randomized self-report designs. Future research will benefit from using a longitudinal design, continued characterization of the workplace bullying-family functioning relationship, including its nature, direction, processes, and boundary conditions in various industrial and cultural contexts, together with the use of models for the integration of research findings.

## Introduction

Workplace bullying is an unethical phenomenon that involves a spectrum of subtle or overtly hostile psychosocial behaviors [1]. Approximately 15% of workers throughout the world experience workplace bullying, making it a global problem that affects several types of occupations and workers [2, 3]. For instance, while the American Workplace Bullying Survey reported that approximately one-third of workers have experienced workplace abuse either directly or indirectly [4], the prevalence of bullying among Chinese employees in Hong Kong and perioperative nurses in Australia was 58.9% and 61%, respectively [5, 6]. Workplace bullying may lead to the target suffering from anxiety [7, 8], depression [7, 9], emotional exhaustion [10, 11], and higher levels of burnout [12, 13]. Additionally, workplace bullying can negatively affect

**Competing interests:** The authors have declared that no competing interests exist.

employees' job performance [14, 15], work engagement [16], job satisfaction [17, 18], organizational commitment [19], and occupational well-being [20], as well as increasing the turnover rate [21, 22].

While a substantial body of evidence demonstrates the detrimental impact of workplace bullying on employee performance in the work sphere, there is limited information on its effects on the family environment. The spillover theory proposes that work and family function as mutually influencing domains, resulting in similarities in terms of emotions, values, skills, and overt behaviors [23]. In addition, the work-family border theory suggests three types of borders between work and family, namely, physical, temporal, and psychological. Permeability between the borders allows the spillover of negative emotions and attitudes from the work sphere into the home sphere [24]. Research indicates that negative experiences at work may harm employees' family functioning [25] and that the quality of family functioning directly impacts the mental health and well-being of family members [26]. As a result, researchers have begun to explore the relationship between workplace bullying and the family outcomes. Tepper et al. reported an association between abusive supervision and greater work-family conflict (WFC) among subordinates and suggested that workplace bullying may permeate the employees' family lives [27]. Abusive supervision can adversely affect the functioning and satisfaction of the family environment [25]. Moreover, supervisors who were undermined in their families during childhood were more likely to engage in abusive supervision in adulthood, highlighting the impact of poor family functioning on workplace behaviors [28].

Despite available evidence on the relationship between family functioning and workplace bullying, the association has not been considered in previous reviews conducted on workplace behaviors [9–11]. For instance, one of the earliest reviews focused on the types of workplace bullying, its individual and organizational impacts, and the role of human resource management in addressing the issue [29]. Fast forward to 2020, Gupta and colleagues conducted a narrative review to classify recent workplace bullying studies and measures taken at the organizational level to address the issue [30]. A recent systematic review looked at the moderators of the relationship between employees' well-being and workplace bullying based on the job demand resources model [31]. Likewise, Özer and Escartín [32] synthesized the empirical studies reporting antecedents, moderators, mediators, and outcomes of workplace bullying from the perpetrators' perspectives. Thus, three types of research gaps can be identified in previous workplace bullying reviews. First, some of the reviews did not employ any systematic approach, which makes the results unreproducible. Second, despite attempts to elucidate the underlying mechanisms between workplace bullying and individual or organizational factors, none of the reviews focused on how workplace bullying affects family functioning. Third, theoretical backgrounds for explaining the factors associated with workplace bullying were not comprehensively reviewed, apart from the job demand resources model explored by Farley et al. [31].

Based on the above-mentioned research gaps, a systematic review offers the opportunity to elucidate the relationship between workplace behaviors and family functioning. Moreover, research on the association between the variables, especially in terms of directions, mechanisms, and boundary conditions, is fragmented and requires further integration. Thus, this systematic review aims to achieve the following objectives (a) present an overview of current research and the direction of the association between workplace bullying and family functioning, and (b) elucidate the mechanisms and moderating factors underlying the relationship between workplace bullying and family functioning. By systematic integration of these studies, we seek a more profound and comprehensive understanding of the field, offering insights for future research endeavors.

## Methods

### Databases and search terms

Four databases were searched for articles, namely, PubMed, PsycINFO, Scopus, and the Web of Science. The searches were conducted in May 2024. All articles published up to May 12, 2024, were targeted. The keywords used for the search were selected before the formal search based on relevant previous studies. Three combinations of keywords were used for searches. First, search terms relating to "work" included "job," "occupation," "employee," and "worker." The second set of keywords was associated with workplace bullying, and included "mobbing," "victimization," and "emotional abuse." The last group of keywords was related to family functioning, for example, "family function*," "family adaptability," and "family relation*." After selecting the search terms, the search strategy was adjusted appropriately according to the requirements of each database. The searches included all studies published on bullying in the workplace and its effect on family functioning. The search items were combined using the Boolean operators "OR" and "AND" are shown in the S1 Table. Databases such as Google Scholar were not searched because of their incompatibility with complex search strings with Boolean operators, the large volume of results from preliminary searches, and the absence of effective filters to identify highly relevant literature.

### Criteria for inclusion and exclusion

Several inclusion criteria were used for the systematic review of the literature to refine the search and identify relevant articles as shown in Table 1. Only original research articles were included to ensure a higher standard of literature in the study. Although we did not restrict the study location or country, non-English articles were excluded, as English-language literature encompasses a broad range of international research findings and includes the most significant studies. Focusing on English-language sources ensures access to high-quality, relevant research materials and enhances screening efficiency while reducing the risk of translation-related errors.

### Data extraction

Searches of the PubMed, PsycINFO, Scopus, and the Web of Science databases for research published up to May 12, 2024, identified a total of 1347 papers. Following independent screening by two authors and the removal of duplicates (n = 223),1124 articles were eligible for title and abstract screening. Thereafter, 1040 articles were excluded for not focusing on workplace bullying and/or family functioning, with 59 excluded based on the abstract and 981 based on the title. After extensive reading and evaluation of the full texts of 84 studies that were deemed

**Table 1. Inclusion and exclusion criteria.**

| Criteria | Included | Excluded |
|---|---|---|
| Aims and objectives | Workplace bullying and family functioning | ① Focused only on either workplace bullying or family functioning or none of the variables |
| | | ② Studies focusing on aspects that may be interpreted differently in various cultural and legal contexts and are distinct from the broader bullying phenomenon |
| Study population | Working populations | Students or non-working populations |
| Study design and type of publications | Quantitative and peer-reviewed | Qualitative, case reports, theoretical papers, review papers, meta-analyses, systematic reviews, conference papers, book chapters, reports, and gray literature |
| Language | English | Non-English written articles |

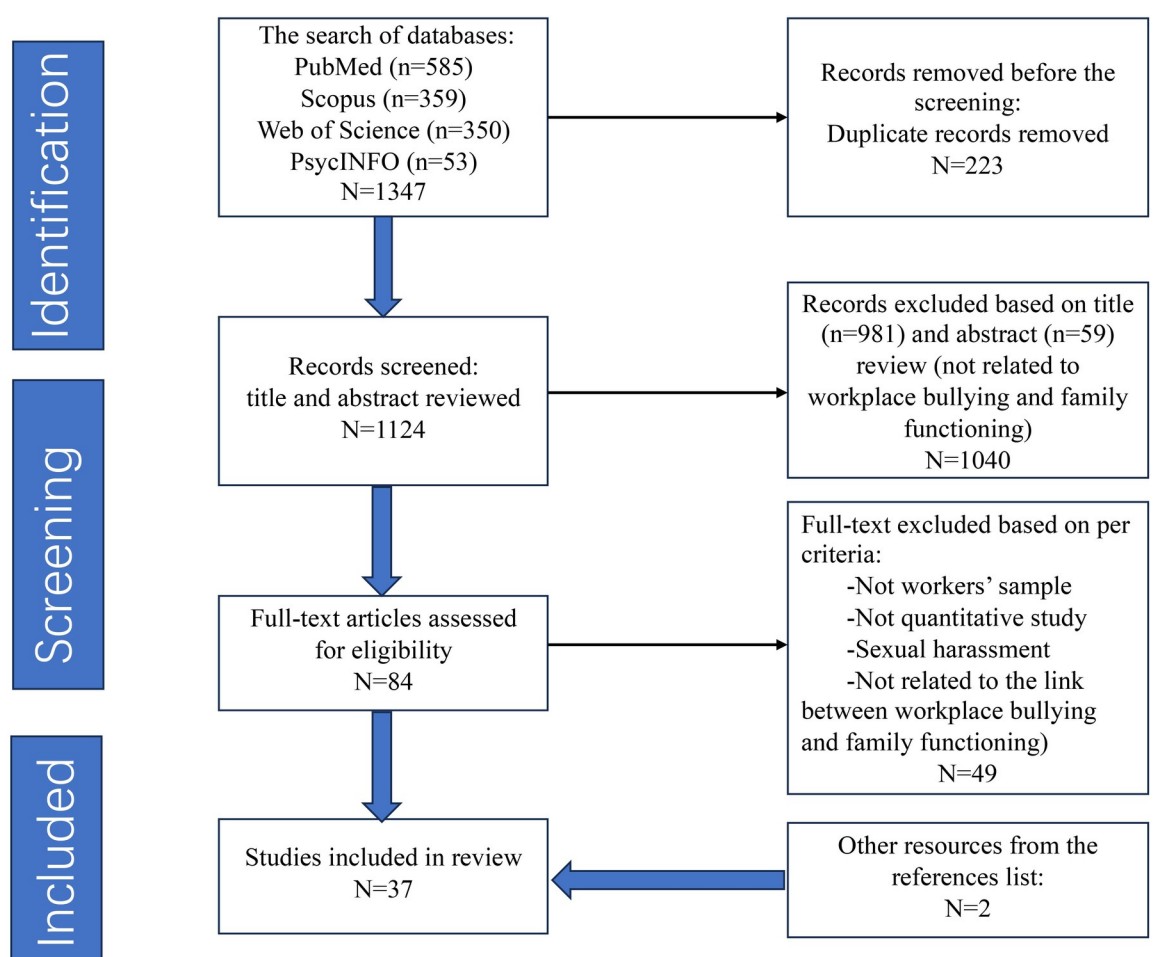

**Fig 1. Flowchart of study retrieval and selection (PRISMA).**

acceptable, studies involving non-working population samples, qualitative studies, theoretical studies, meta-analyses, systematic reviews, and studies on sexual harassment, racial harassment, and studies that did not address the connection between workplace bullying and family functioning (n = 49) were excluded. In addition, the reference lists of articles were searched for relevant studies, yielding two articles that qualified for inclusion. Finally, 37 studies were considered eligible for inclusion in the review. We made attempts to retrieve unavailable full texts by sending a formal request to the corresponding author of the article. Reminders were sent twice after which such articles were excluded if no response was received from the corresponding author. Fig 1 illustrates the screening procedures for studies.

## Quality assessment tool

Quality assessments were conducted independently by the first and corresponding authors by using a revised version of a previously established methodological quality checklist [33]. The original checklist comprised 13 items which were adapted appropriately to suit the content of the review. For instance, the original item "How was sickness absence assessed?" was modified to "How was family functioning assessed?" and the response options to "Self-report-0; Data from the spouses, family member-1" were adjusted to align with the focus of the study.

Additionally, irrelevant questions, such as "Was previous sickness absence adjusted for in prospective analyses?" were removed. After these revisions, the quality checklist consisted of 10 items that provided information such as sampling, response rate, representativeness, selection bias, measurement, sample size, and the presence of confounders. The revised methodological quality checklist is provided in the S2 Table. The checklist scores ranged from 0 (lowest study quality) to 10 (highest study quality). Scores between 7 and 10 were considered high quality, with low quality indicated by scores below 3, and moderate quality by scores between 4 and 6. Disagreements regarding study quality were resolved by discussion.

## Results

### Study characteristics

Table 2 summarizes the features of the 37 included papers. The studies came from North America (N = 11), Asia (N = 22), South America (N = 1), Africa (N = 1), Europe (N = 1), Oceania (N = 1) and were published between 2006 and 2024. Among them,1 study was published in 2006, and 36 were published after 2010. In terms of the sample distribution, four (10.81%) articles did not report the sample industry studied, while 15 (40.54%) included samples from various industries, with the 16 samples (43.24%) obtained from specific industries, the healthcare, hospitality, manufacturing, construction, and sales sectors. Additionally, 2 articles (5.41%) utilized samples from both specific and multiple industries concurrently.

Second, most of the included studies employed cross-sectional, non-randomized self-report designs. Of the 37 articles, 17 were cross-sectional, 17 were time-lag studies, and 3 used experience sampling (diary studies).

Of the 37 studies, 13 examined bullying using the Abusive Supervision Scale (ASS). This consists of the original 15-item ASS [27], the revised ASS [66], and the shortened ASS [27, 67]. 10 studies used the Negative Acts Questionnaire (NAQ), which includes the original NAQ [68] and the Revised NAQ [69]. Six studies utilized the Workplace Ostracism Scale developed by Ferris [70] and its simplified version to measure workplace ostracism. Other studies used several instruments, including the Negative Workplace Gossip Scale [71] and the Workplace Aggression Research Questionnaire [72], to measure the prevalence of workplace gossip and aggression, respectively.

In terms of quality assessment, the inter-rater agreement for study quality was 81.08%, and Cohen's Kappa demonstrated substantial agreement between the raters (k = 0.700). The quality ratings for the included studies ranged from 4 to 9, with a mean value of 6.92 (SD = 1.057). Most of the examined studies were cross-sectional in design, with reported response rates of over 50% and employed non-probability sampling methods.

Together with self-reported measures, family member ratings were used to assess family functioning. Furthermore, most studies diligently controlled for variables that previous research has indicated as potential factors influencing workplace bullying and family functioning. These variables encompassed demographic factors (e.g., age, gender, education, and marital status), work-related variables (e.g., tenure, work hours, work performance, number of employees), and family-related factors (e.g., number of children living at home and the length of the marital relationship). The inclusion of these control variables allowed the researchers to eliminate the influence of other potential factors on the study results, ensuring accuracy and reliability.

### Workplace bullying and its impact on family functioning

Five indicators were selected for the assessment of family functioning in the 37 included studies. These indicators were (a) work-family conflict (WFC), (b) family undermining, (c) family

**Table 2. Overview of included studies.**

| Reference (year) | Region | Study-design | Theories | Industry | N[a] | Multi-resource data | Sample type | Relationship examined | Quality Score[d] | |
|---|---|---|---|---|---|---|---|---|---|---|
| Hoobler et al. (2013) [34] | North America | Cross-sectional | "Trickle-down effects" of abusive supervision | Various | 200 tri-matched | Yes | Non-random | WB[b]→FF[c] | 8 | 8 |
| Barber et al. (2017) [35] | North America | Cross-sectional | Self-regulation theory | Various | 118 pairs | Yes | Non-random | WB→FF | 7 | 6 |
| Raja et al. (2018) [12] | South Asia | Cross-sectional with time-lag | COR theory[e] | Government organizations | 151 | No | Non-random | WB→FF | 5 | 5 |
| Hoobler et al. (2006) [36] | North America | Cross-sectional | Displaced aggression theory | Various | 210 tri-matched | Yes | Non-random | WB→FF | 8 | 8 |
| Pradhan et al. (2021) [37] | South Asia | Cross-sectional | COR theory Crossover theory | Various | 188 pairs | Yes | Non-random | WB→FF | 8 | 8 |
| Carlson et al. (2012) [38] | North America | Cross-sectional with time-lag | COR theory | Various | 328 | No | Non-random | WB→FF | 6 | 7 |
| Ju et al. (2020) [39] | East Asia | Cross-sectional | The work-family interface model | Construction | Study1:361 Study 2:608 | No | Non-random | WB→FF | 6 | 6 |
| Viotti et al. (2018) [40] | North America | Cross-sectional | COR theory | Nurses | 333 | No | Non-random | WB→FF | 7 | 7 |
| Sarwar et al. (2023) [41] | South-east Asia | Cross-sectional with time-lag | The work-family interface model | Service (restaurants) | 238 | No | Non-random | WB→FF | 6 | 6 |
| Selem et al.2024 [42] | Africa | Cross-sectional with time-lag | Affective events theory | Service (hotel) | 519 | No | Non-random | WB→FF | 6 | 7 |
| Zhou et al. (2018) [43] | North America | Cross-sectional | The stress/strain framework | Various | 2058 | No | Random | WB→FF | 7 | 7 |
| Pluut et al. (2022) [44] | East Asia | Experience sampling study | Spillover theory COR theory | Not report | 70 pairs | Yes | Non-random | WB→FF | 7 | 9 |
| Liang (2020) [45] | East Asia | Cross-sectional with time-lag | Affective events theory Crossover theory | Plastics corporation | 569 pairs | Yes | Non-random | WB→FF | 8 | 8 |
| Yoo et al. (2018) [20] | East Asia | Cross-sectional | Work-family interface model | Various | 307 | No | Non-random | WB→FF | 7 | 7 |
| Zhang et al. (2017) [46] | East Asia | Cross-sectional with time-lag | COR theory | Manufacturing | 323 pairs | Yes | Non-random | WB→FF | 8 | 8 |
| Chi et al. (2018) [47] | East Asia | Experience sampling study | Stressor-emotion model | Various | 77 | No | Non-random | WB→FF | 7 | 7 |
| Choi et al. (2019) [48] | East Asia | Cross-sectional | The transactional stress model | Various | 285 | No | Non-random | WB→FF | 4 | 4 |
| Sarwar et al. (2021) [49] | South Asia | Cross-sectional with time-lag | Spillover theory | Nurse | 251 | No | Non-random Random | WB→FF | 7 | 7 |
| Machado et al. (2021) [7] | South America | Cross-sectional | Resource drain theory | Service (gastronomy) | 160 | No | Non-random | WB→FF | 6 | 6 |
| Valle et al. (2021) [50] | North America | Cross-sectional with time-lag | COR theory | Not report | 260 | No | Non-random | WB→FF | 6 | 6 |
| Thompson et al. (2020) [51] | North America | Cross-sectional with time-lag | W-HR model[f] Crossover theory | Not report | 350 pairs | Yes | Non-random | WB→FF | 7 | 9 |
| Carlson et.al (2011) [25] | North America | Cross-sectional | Spillover theory Crossover theory | Various | 280 pairs | Yes | Non-random | WB→FF | 7 | 7 |
| Choi et al. (2018) [52] | East Asia | Cross-sectional | The transactional stress model | Various | 310 | No | Non-random | WB→FF | 5 | 5 |
| Chen (2018) [53] | East Asia | Cross-sectional | Stressor–strain model Spillover theory | Service | 457 pairs | Yes | Non-random | WB→FF | 8 | 8 |
| Zhu et al. (2023) [54] | East Asia | Cross-sectional with time-lag | Ego depletion theory Crossover theory | Service | 230 pairs | Yes | Non-random | WB→FF | 7 | 8 |
| Zhang et al. (2019) [55] | East Asia | Cross-sectional with time-lag | Stressor-strain-outcome COR theory | Service | 221 | No | Non-random Random | WB→FF | 8 | 8 |
| Dionisi et al. (2019) [56] | North America | Cross-sectional | COR theory | Not report | 123 pairs | Yes | Non-random | FF→WB | 7 | 7 |
| Restubog et al. (2011) [57] | Oceania | Cross-sectional with time-lag | The transactional stress model | Various | Study1:184 tri-matched Study2:188 pairs | Yes | Non-random | WB→FF[c] | 8 | 8 |
| Choi (2021) [58] | East Asia | Cross-sectional | COR theory | Various | 226 | No | Non-random | WB→FF | 6 | 6 |
| Wu et al. (2012) [59] | East Asia | Cross-sectional with time-lag | Work-family interface model | Manufacturing | 209 | No | Non-random Random | WB→FF | 8 | 8 |
| Liu et al. (2013) [60] | East Asia | Cross-sectional with time-lag | Work-family interface model | Manufacturing | 233 | No | Non-random Random | WB→FF | 7 | 6 |
| Kiewitz et al. (2012) [28] | South-east Asia | Cross-sectional | Social learning theory | Various | Study1:179 pairs Study2:97 tri-matched | Yes | Non-random | FF→WB | 7 | 7 |
| Liu et al. (2022) [61] | East Asia | Cross-sectional with time-lag | Spillover theory | Service (hotel) | 286 | No | Non-random | WB→FF | 7 | 7 |
| Rodríguez-Muñoz et al. (2017) [62] | Europe | Experience sampling | COR theory | Various | 68 pairs | Yes | Non-random | WB→FF | 8 | 8 |
| Li et al. (2024) [63] | East Asia | Cross-sectional with time-lag | W-HR model | Various | 431 | No | Non-random | WB→FF | 7 | 7 |
| Rafique et al. (2024) [64] | South Asia | Cross-sectional with time-lag | COR theory | Construction | 235 | No | Non-random | FF→WB | 6 | 5 |

*(Continued)*

**Table 2.** (Continued)

| Reference (year) | Region | Study-design | Theories | Industry | Nᵃ | Multi-resource data | Sample type | Relationship examined | Quality Scoreᵈ | |
|---|---|---|---|---|---|---|---|---|---|---|
| Demsky et al. (2014) [65] | North America | Cross-sectional | COR theory | Various | 107 tri-matched | Yes | Non-random | WB→FF | 7 | 7 |

Note

ᵃ "N" represents the final sample.

ᵇ "WB" represents workplace bullying.

ᶜ "FF" represents family functioning.

ᵈ Methodological quality score (scale: 0–10).

ᵉ "COR theory" represents the conversation of resources theory.

ᶠ "W-HR model" represents the work-home resources model.

satisfaction, (d) marital behaviors, and (e) family emotional exhaustion. Figs 2 and 3 illustrate the mediators, moderators, and theories that linked workplace bullying to family functioning, with parentheses indicating the number of tests conducted for each variable.

**Workplace bullying: Work-family conflict (WFC) relationships.** Eighteen studies examined the correlation between workplace bullying and WFC, and two concentrated on workplace bullying and work-family interference (WFI). Of the studies on WFC, five explored the direct relationship, three analyzed the indirect connection, and 10 investigated both direct and indirect associations. All studies showed a direct or indirect positive correlation between bullying and WFC. Moreover, two studies on WFI yielded consistent findings. Workplace bullying positively correlates with WFI [7] and indirectly influences strain-based and time-based interferences from work to private life [40].

*Mechanisms linking workplace bullying and WFC.* Fourteen studies explored the mechanisms linking workplace bullying to WFC, consistently supporting an indirect effect. The

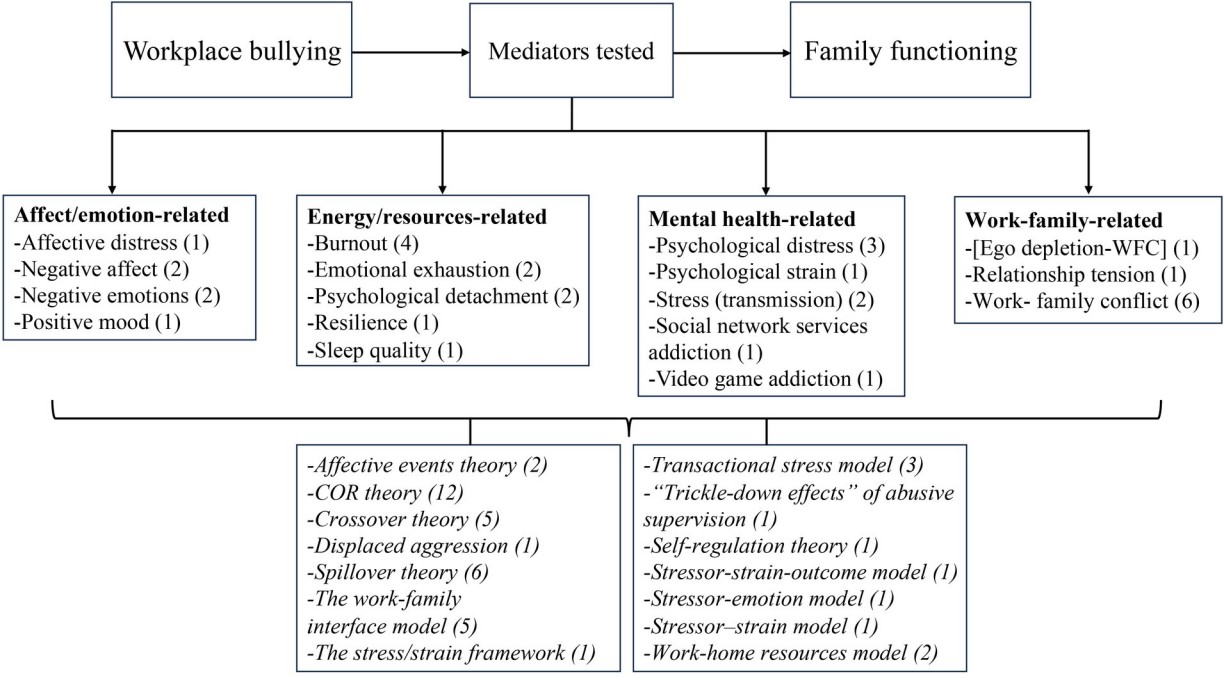

**Fig 2. Mediators organized by category.**

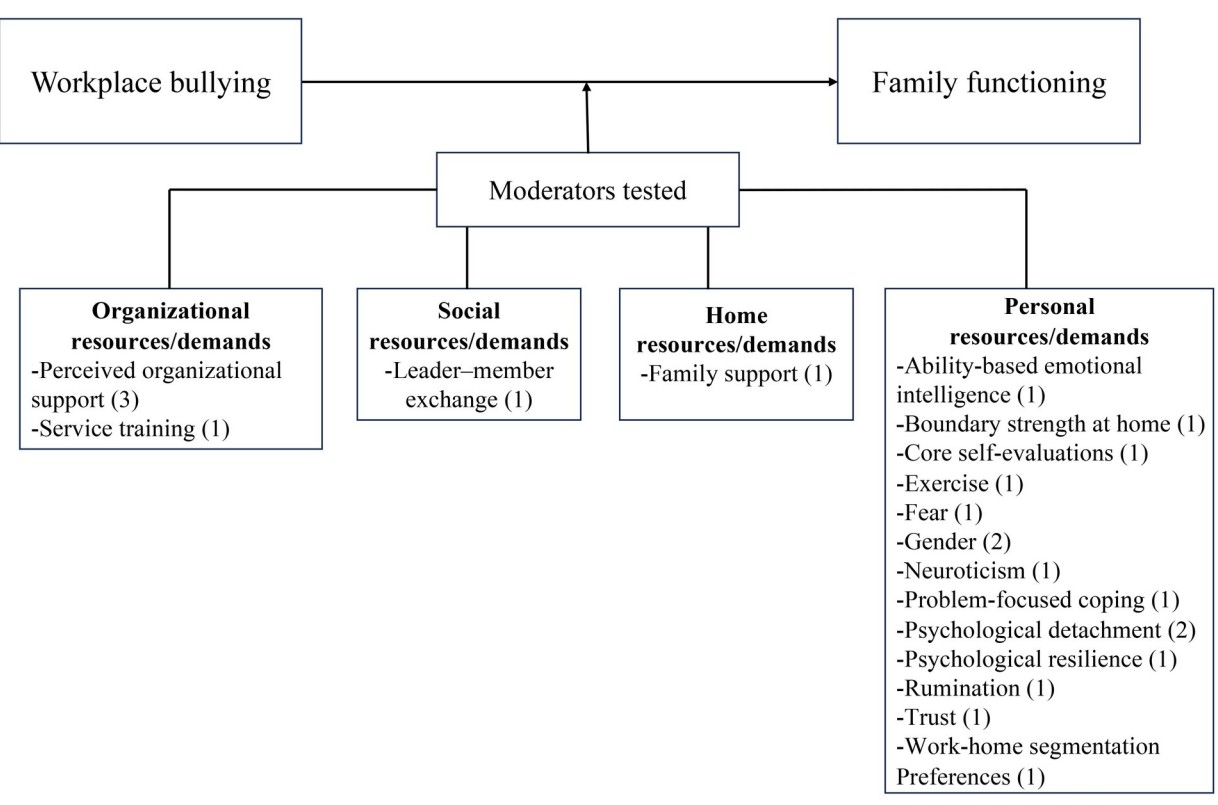

**Fig 3. Moderators organized by category.**

mediating factors can be classified broadly into affect (emotion)-related, energy (resources)-related, and mental health-related. These various mediators were investigated based on different theories, with burnout being the most extensively studied, followed by negative affect or emotions (Fig 4).

*Moderators between workplace bullying and WFC.* Eleven moderation tests offered insights into the boundary conditions of the relationship between workplace bullying and WFC. According to Lee et al.'s [73] taxonomy of resources and demands, these moderators encompassed three critical areas, namely, organizational, home, and personal resources (demands) (Fig 5).

Three moderation tests concerning organizational resources (demands), and all were significant. Perceived organizational support (POS) alleviated the indirect effects of workplace bullying on WFC through video game addiction [52]. Workplace ostracism showed a stronger positive association with WFC among female employees with low POS [58]. Customer service training also alleviated the detrimental impacts of daily customer mistreatment on service employees' daily WFC [47].

Regarding home resources (demands), only one moderation test was reported, with results indicating that family support moderated the indirect effect of abusive supervision on WFC via psychological distress [39]. Family support played distinct roles at different stages: it intensified the impact of abusive supervision on psychological distress but mitigated the effects of psychological distress on WFC.

Seven moderation tests were tested within the personal resources (demands) category, of which five were significant. Two significant tests exhibited exacerbating effects, while three showed mitigation effects. Moderators in this category covered various individual-level

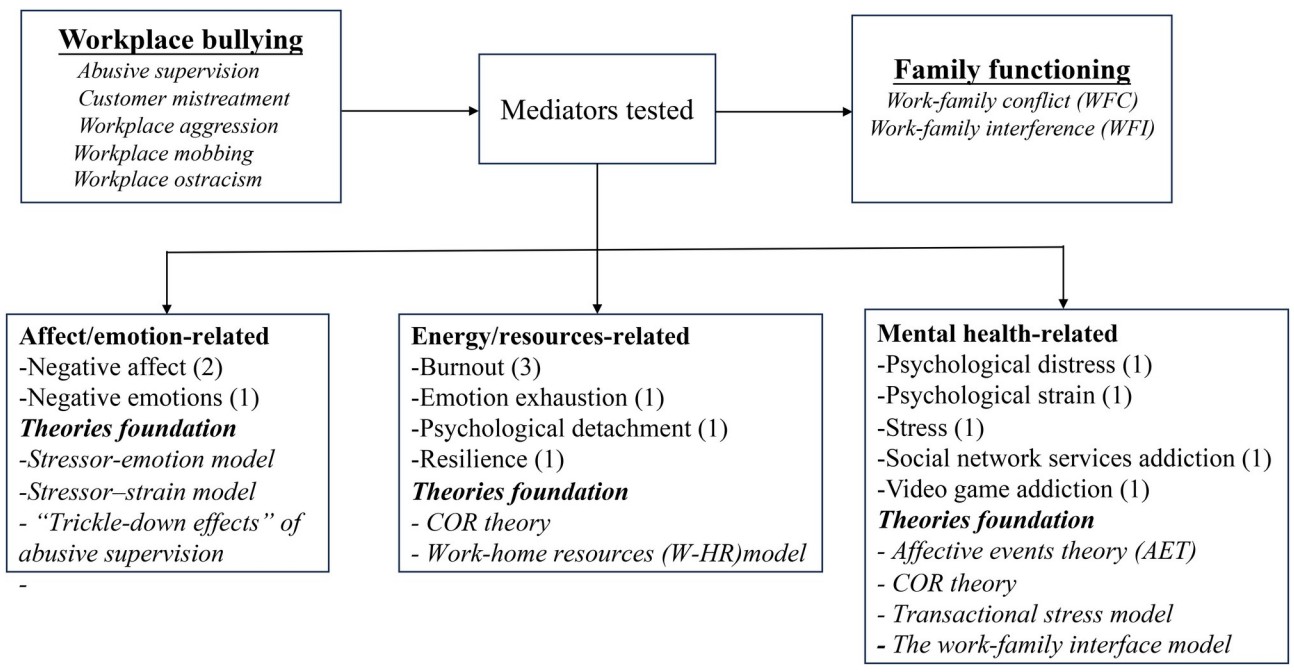

**Fig 4. Mediators and theories associated with the relationship between workplace bullying and WFC.**

characteristics, including demographics, coping strategies, psychological states, and personal traits. Specifically, fear and rumination significantly intensified the indirect relationship between workplace ostracism and WFC [41, 46]. Two personal traits, core self-evaluation, and

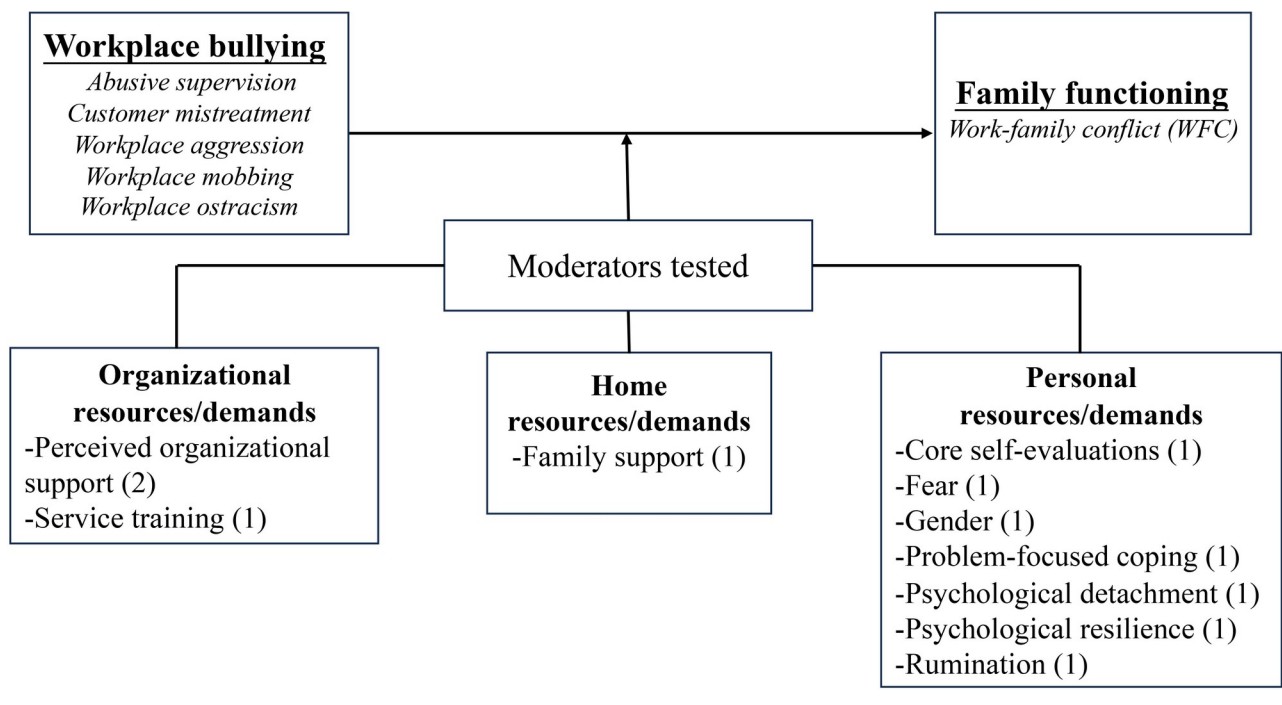

**Fig 5. Moderators of the association between workplace bullying and WFC.**

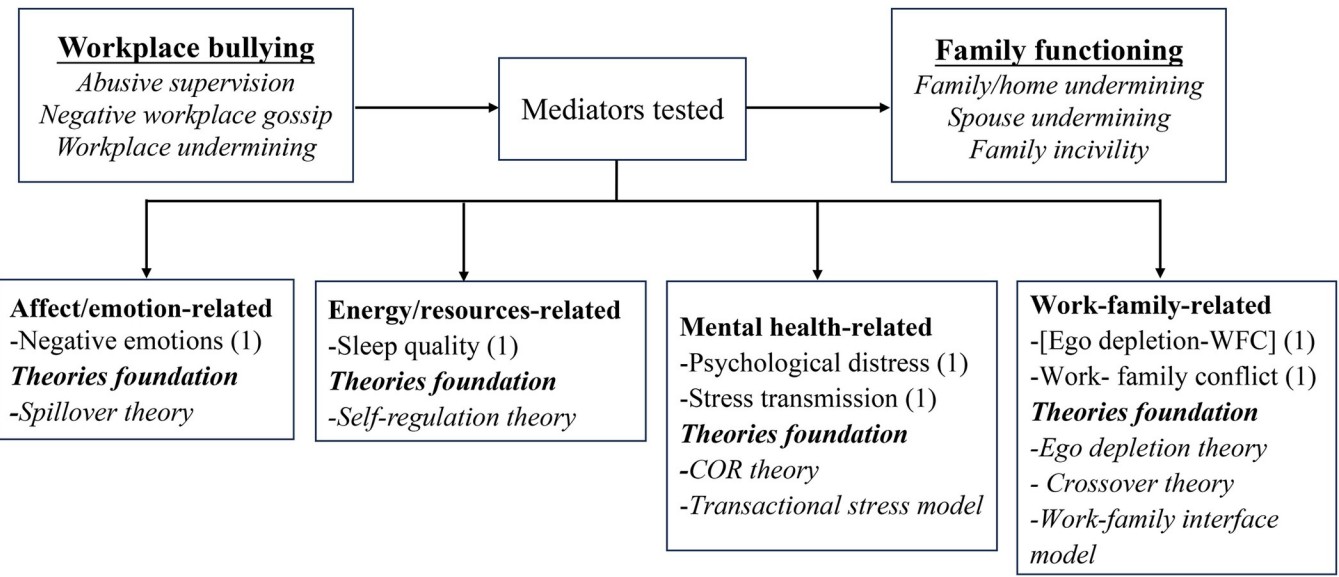

**Fig 6. Mediators and theories associated with the relationship between workplace bullying and family undermining.**

psychological resilience, depicted significant mitigating effects [47, 63]. Meanwhile, high levels of problem-focused coping reduced the impact of supervisor aggression on negative emotions [53]. However, the moderating role of psychological detachment in the relationship between workplace ostracism and WFC in a group context was not supported [63], and no gender differences exist in the impact of abusive supervision on WFC [43].

**Workplace bullying: Undermining relationships in the family.** Seven studies examined the impact of workplace bullying on family undermining. One study analyzed the direct relationship between bullying and family undermining; four studies explored its indirect effects; and two studies investigated both direct and indirect relationships. These studies consistently supported that workplace bullying is positively associated with family undermining, both directly and indirectly.

*Mechanisms associated with the link between workplace bullying and family undermining.* Drawing upon various theoretical frameworks, researchers examined six mediators linked to emotions, energy, mental health, and work-family relationships (Fig 6). Across all studies, consistent findings indicated the existence of indirect effects.

*Moderators between workplace bullying and family undermining.* Researchers conducted five moderation tests exploring the boundary conditions between workplace bullying and family undermining, all of which yielded significant results. These examinations indicated that organizational and personal resources played a significant moderating role in this relationship (Fig 7).

Within the organizational category, perceived organizational support (POS), viewed as a vital contextual resource, mitigated the indirect link between negative workplace gossip and family undermining [54]. Personal resources like boundary strength at home and exercise also showed similar buffering effects [35, 59]. Moreover, gender moderated the relationship, with men experiencing abusive supervision showing a stronger link between psychological distress and spouse undermining [57]. Additionally, neuroticism worsened the connection between negative emotions caused by workplace bullying and family incivility [49].

**Workplace bullying: Family satisfaction.** Four studies investigated the impact of workplace bullying on family satisfaction. Two focused on indirect relationships, while the other

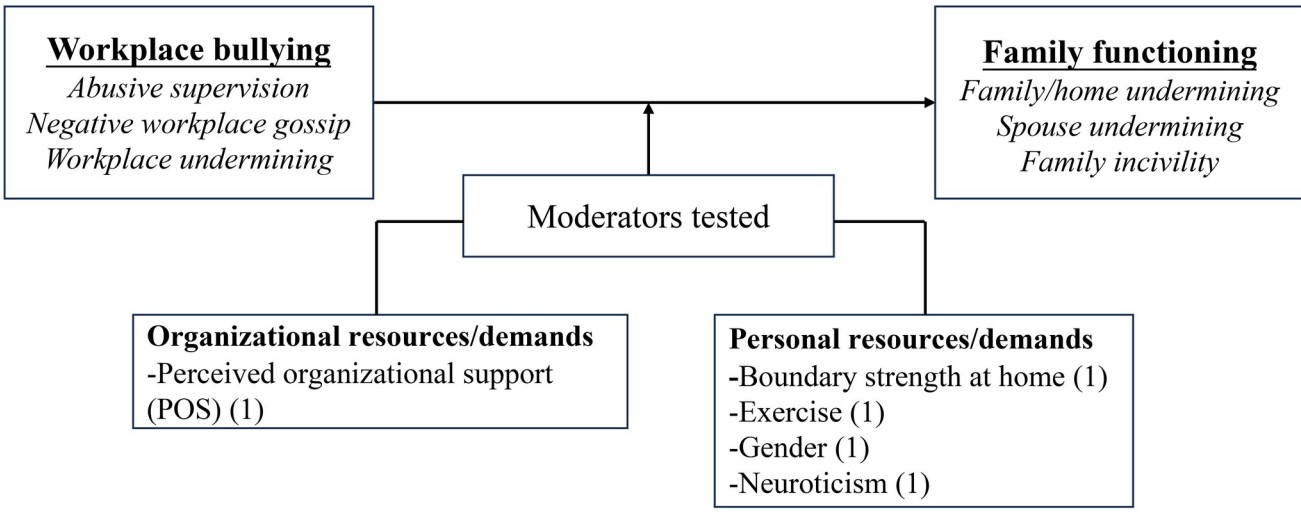

**Fig 7. Moderators associated with the relationship between workplace bullying and family undermining.**

two examined both direct and indirect relationships. All studies indicated that workplace bullying is associated with family satisfaction, either directly or indirectly.

*Mechanisms associated with the link between workplace bullying and family satisfaction.* Two mediators related to the work-family relationship were investigated in the connection between workplace bullying and family satisfaction. Three studies demonstrated that work-family conflict (WFC) is a critical mechanism linking workplace bullying to family satisfaction, while one study did not support this finding [25]. Additionally, relationship tension was confirmed as an important mediator [25]. These investigations primarily draw on perspectives from spillover theory, COR theory, crossover theory, the stress-strain-outcome model, and the work-family interface model. (Fig 8).

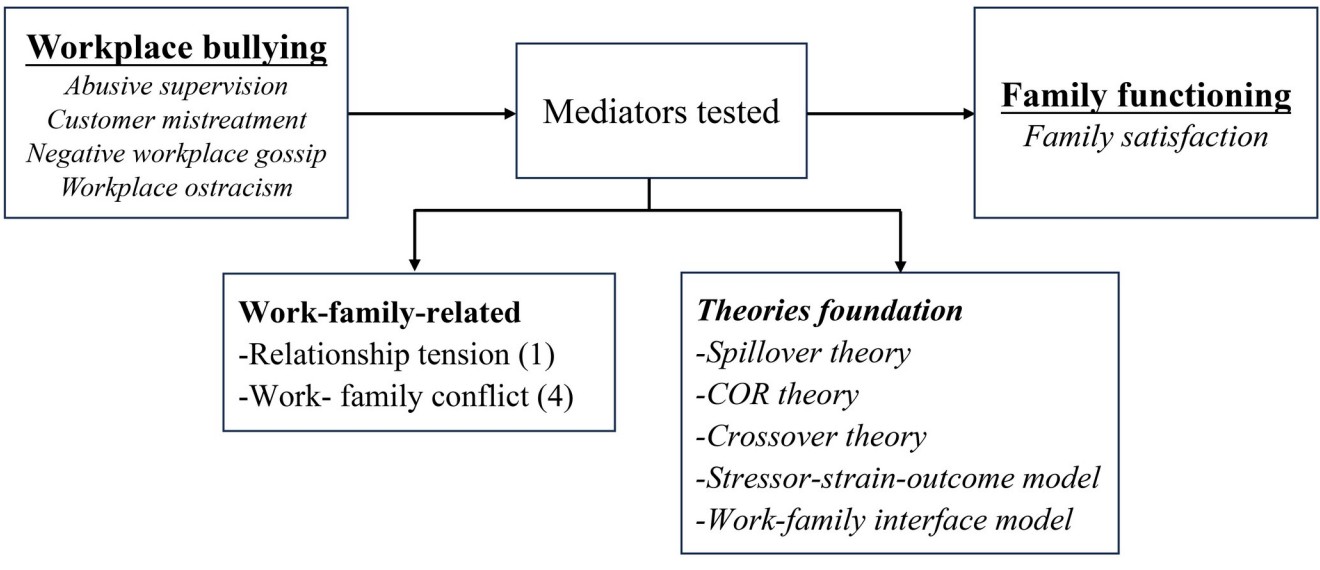

**Fig 8. Mediators and theories: Workplace bullying and family satisfaction.**

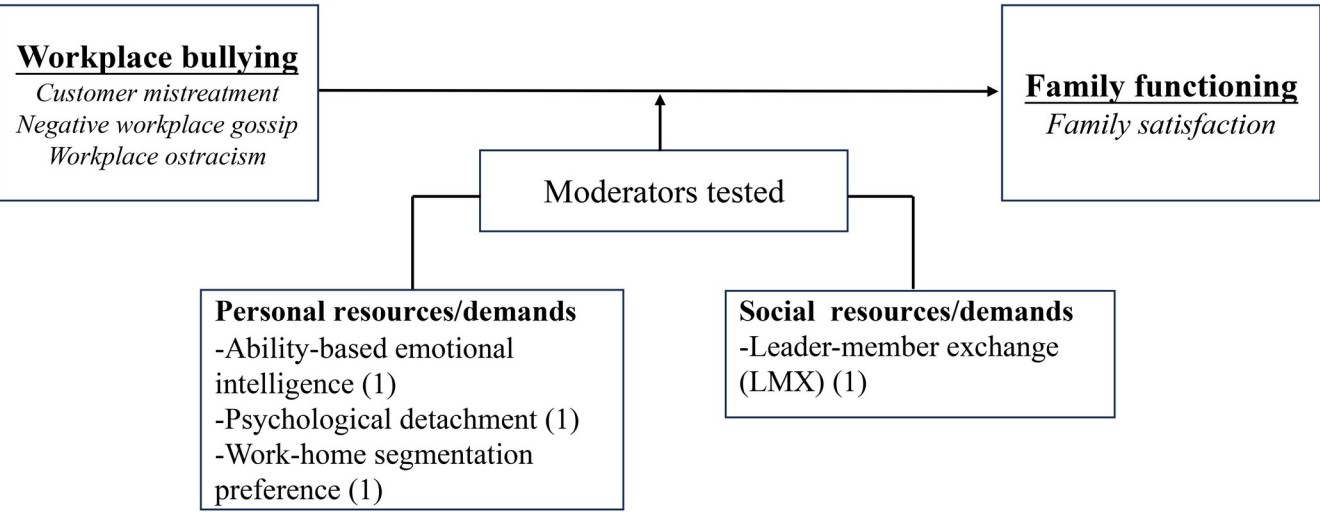

**Fig 9. Moderators of the relationship between workplace bullying and family satisfaction.**

*Moderators of the relationship between workplace bullying and family satisfaction.* There are four moderation tests, primarily focusing on social and personal resources. All moderation tests yielded significant results, demonstrating an attenuating effect (Fig 9). Notably, Leader-member exchange (LMX), as a social resource, significantly reduced the direct impact of customer mistreatment on WFC and the indirect influence on family satisfaction [55]. Higher levels of emotional intelligence weakened the impact of negative workplace gossip on family satisfaction [61]. Moreover, increased psychological detachment mitigated the adverse effects of customer mistreatment on family satisfaction via WFC [55]. Furthermore, high work-home segmentation preferences alleviated the impact of workplace ostracism on WFC [60].

**Workplace bullying: Marital behaviors and relationships.** Two experience-sampling studies explored the repercussions of workplace bullying on marriages. One study found that social stressors, such as abusive supervision, indirectly led to anger, withdrawal, and reduced supportive marital behaviors [44]. Another suggested an indirect correlation between workplace bullying and home conflicts reported by employees and their spouses [62]. Furthermore, the study also revealed that despite the lack of a direct association between bullying and relationship satisfaction, it indirectly influenced self-reported relationship satisfaction through affective distress and psychological detachment [62].

*Mechanisms associated with the link between workplace bullying and marital behaviors.* Two diary studies explored the mediating role of three factors related to affect and resources in the relationship between workplace bullying and marital behaviors and confirmed their significance. The theories for these findings lie in the spillover and crossover theory (Fig 10).

*Moderators between workplace bullying and marital behaviors.* Only one study examined trust as a moderator. Pluut et al. found that trust significantly aggravated the impact of social stressors on burnout. Individuals with higher levels of trust may be more susceptible to the experience of burnout when facing abusive supervision [44].

**Workplace bullying: Emotional exhaustion in the family.** Only one study explored the impact of workplace ostracism on family emotional exhaustion among employees and their spouses. Applying the work-home resources (W-HR) model, the study proposed two pathways: workplace ostracism affects family emotional exhaustion by diminishing positive mood and increasing psychological distress. The results indicated that both employees' and spouses'

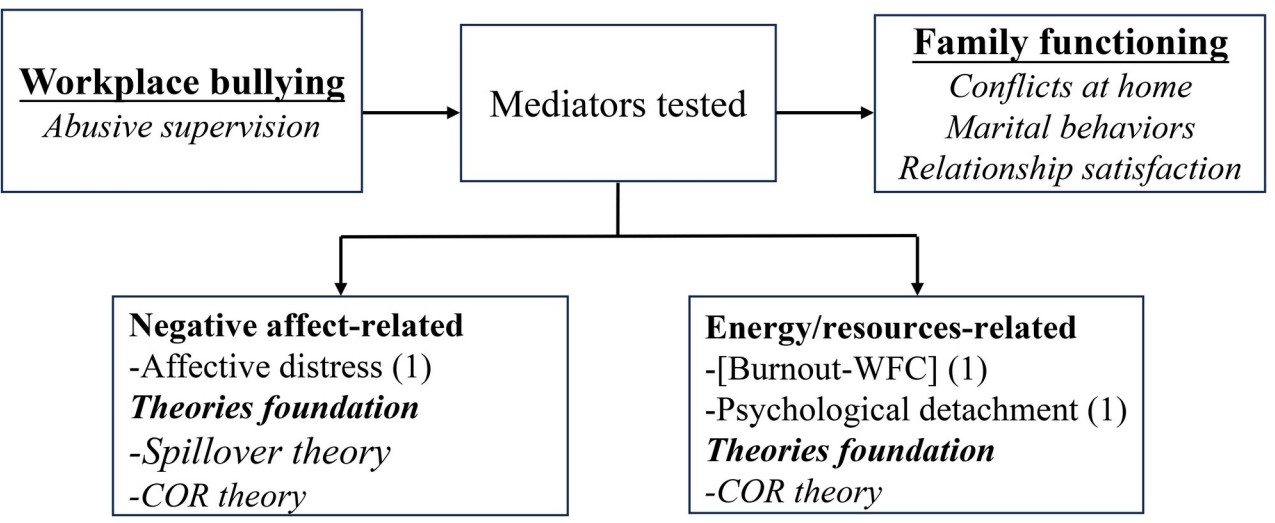

**Fig 10. Mediators and theories: Workplace bullying and marital behaviors.**

family emotional exhaustion primarily arose through psychological distress rather than positive mood [51].

### Family functioning and its impact on workplace bullying

Three studies (8.11%) investigated the influence of family functioning on workplace behaviors and highlighted the importance of family functioning in shaping workplace bullying. Specifically, supervisors' experiences of family undermining were positively associated with perceived abusive supervision of their subordinates [28]. Time-based, strain-based, and behavior-based WFC are positively related to abusive supervision [64]. Furthermore, research has shown the indirect effects of WFC and romantic relationship conflict on abusive supervision [56, 64]. All findings support the direct or indirect effects of family dynamics and workplace behaviors, particularly in the context of abusive supervision and workplace bullying (Fig 11).

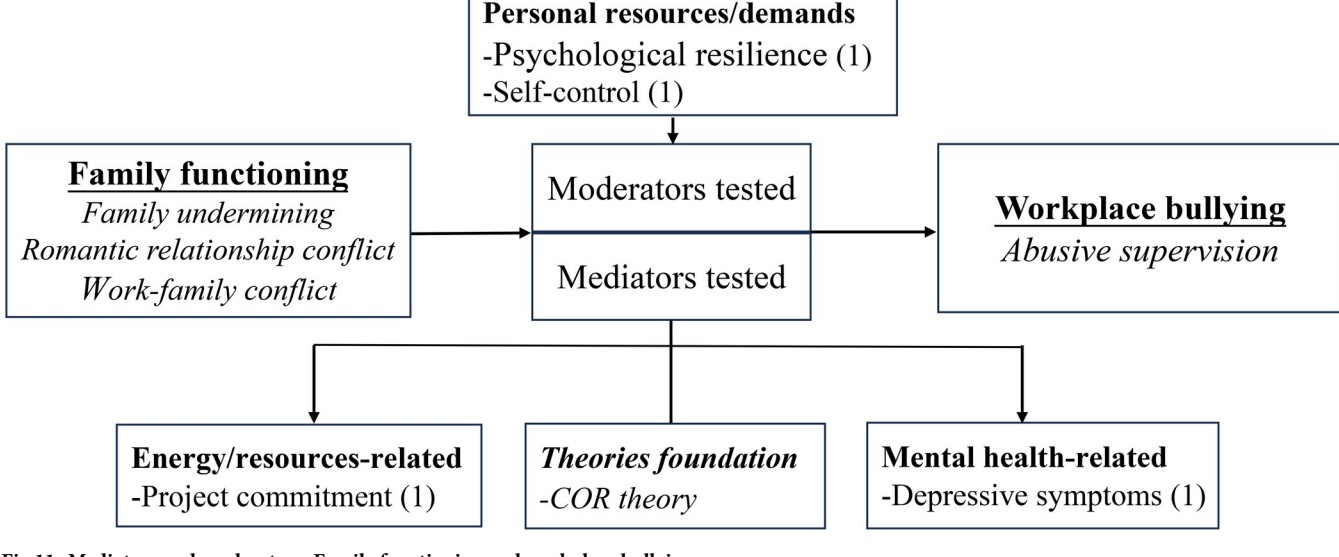

**Fig 11. Mediators and moderators: Family functioning and workplace bullying.**

**Mechanisms associated with the link between family functioning and workplace bullying.** Two studies investigating how family functioning influences workplace bullying highlighted the significance of job commitment and depressive symptoms based on the COR theory (Fig 11).

**Moderators between family functioning and workplace bullying.** Two moderation tests related to personal resources were performed in this context, which were significant and showed alleviating effects (Fig 11). Specifically, higher levels of psychological resilience were associated with weaker adverse effects of WFC on project commitment [64]. Supervisors with high self-control could mitigate the impact of family undermining on abusive supervision by foreseeing the consequences of their actions and thus avoiding such behavior [28].

## Discussion

This systematic review rigorously followed the PRISMA guidelines, identifying 37 quantitative studies on the relationship between workplace bullying and family functioning. Most of the studies were cross-sectional and time-lag research designs, which are expected since the objectives were to identify the relationship between workplace bullying and family functioning. Moreover, the use of longitudinal studies has been recently recommended to elucidate the daily dynamics of bullying and its consequences [74].

### Relationship between workplace bullying and family functioning indicators

This study identified five main indicators of family functioning that were explored in the reviewed studies, which included work-family conflict (WFC), family undermining, family satisfaction, marital behaviors, and family emotional exhaustion. The mediating and moderating factors were also identified, thus providing robust information on the relationships between the indicators and workplace bullying. Our understanding of workplace bullying and family functioning was further supported by the underlying theories. Each of these family functioning indicators are discussed below.

**Work-family conflict (WFC).** WFC arises from incompatible demands between an individual's professional and home lives [75]. Positive correlations were observed between workplace bullying and work-family interference, with variables relating to a negative affect (emotion), energy or resources, and mental health as mediators. These findings were gleaned from several theories such as the COR theory, stressor-emotion and stressor-strain model, work-family interface theory, and transaction stress model.

Negative affect (emotion) is crucial in understanding workplace bullying-WFC relationships. According to the stressor-emotion [47] and stressor-strain models [53], workplace bullying acts as a stressor and triggers negative emotional and affective reactions in the target. Subsequently, the target may carry these negative emotions home, leading to WFC.

Regarding resource-related factors, the examination of burnout, resilience, and psychological detachment was grounded in the COR theory, which posits that people attempt to preserve, develop, and safeguard resources. These resources can be defined as personal traits, conditions, or energies individuals value or allow them to achieve their goals [76]. Within this framework, bullying is considered a significant occupational stressor, as coping with stressors can drain an individual's resources and result in burnout. The drain on resources reduces the ability to deal effectively with family demands, thus contributing to [12, 38].

Mental health variables also provide insights into how workplace bullying affects WFC. Based on the affective events (AET) theory, psychological strain denotes the employee's response to bullying. Employees experiencing psychological strain may carry this burden and negative work experiences into their family life when they leave their organizations,

contributing to difficulties in fulfilling family obligations and resulting in WFC [45]. From the view of the work-family interface model, work and family domains interact with stress in one domain, affecting behaviors and emotions in another.

Abusive supervision leads to psychological distress in employees (e.g., depression and anxiety), and these negative emotions and stresses can spill over from work to home, disrupting family interactions and causing WFC [39]. In addition, drawing upon the transactional stress model, workplace bullying can induce stress in employees, prompting them to turn to social networking services and video games as coping mechanisms. Excessive engagement in these activities and significant time consumption may hinder employees from effectively fulfilling their family responsibilities, intensifying WFC [48, 52]. It was not surprising as factors ranging from organizational resources (i.e., perceived organizational support and service training), to home resources such as family support, and personal resources (i.e., demographics, coping strategies, psychological states, and personal traits) were identified as moderators of the relationship between workplace bullying and WFC.

**Family undermining.**   Family undermining demonstrated a consistent positive association with workplace bullying either directly or indirectly. Family undermining and family incivility involve the initiation of disrespectful behaviors towards family members, adversely affecting the establishment and maintenance of positive interpersonal relationships in the family [49, 59]. Despite their differences in purpose and severity, these types of behaviors involve adverse treatment that can emotionally and psychologically harm family members, potentially leading to strained or broken family relationships. These events explain the exploration of family incivility and undermining as a combined category in the reviewed studies.

From the spillover theory viewpoint, workplace bullying serves as a stress-inducing event and triggers negative emotions in individuals, which could spill over into the home environment and prompt targets to exhibit incivility [49]. Similarly, the transactional stress model suggests that abusive supervision acts as an external stressor and triggers psychological distress in employees. Affected individuals tend to employ emotion-focused coping strategies, transferring negative emotions to family members and exhibiting undermining behaviors to alleviate negative emotions or frustration [57].

Drawing from the COR theory, employees subjected to technology-enacted abusive supervision may invest their resources in surface acting during emotional labor, leading to potential emotional exhaustion. With work stress spilling over into the family domain and the employees being too exhausted to manage it effectively, they might be more likely to express anger towards family members and display undermining behaviors [50]. Moreover, self-regulation theory posits that workplace undermining depletes employees' energy, requiring recovery through rest and sleep. However, poor sleep quality disrupts this recovery process, compromising employees' self-regulation abilities. This impairment makes it difficult for them to manage negative emotions and behaviors at home, thereby increasing home undermining [35].

Finally, WFC reveals the pathways through which workplace bullying impacts family undermining. Grounded in ego depletion and crossover theory, negative workplace gossip depletes employees' self-regulatory resources, resulting in ego depletion. This depletion makes balancing work and family demands challenging, exacerbating WFC. Subsequently, these conflicts and negative emotions are transmitted across domains to the family, leading to negative interactions with partners and undermining behaviors within the family [54]. Moreover, Voydanoff's work-family interface model suggests a link between work characteristics and WFC, which is related to outcomes in the family domain [77]. Thus, abusive supervision exacerbates WFC and subsequently leads to family undermining [59].

**Family satisfaction.**   Another indicator of family functioning identified in this review was family satisfaction–an essential metric of individual well-being [26] and a reflection of a

person's attitude towards family life or the family situation [78]. Abusive supervision exacerbated relationship tension, resulting in a lack of motivation and ability to interact positively with family members, affecting family satisfaction and functioning [25]. These findings highlight the potential diversity and complexity of how workplace bullying affects family satisfaction. Additionally, WFC constitutes a family consequence of bullying and a crucial mediator linking bullying with family satisfaction. Hence, future research may focus on reducing the impact of bullying on WFC.

Overall, factors such as leader-member exchange (LMX), higher levels of emotional intelligence, and increased psychological detachment significantly reduced the negative effects of various events (customer mistreatment, negative workplace gossip, and workplace ostracism) on WFC and their indirect effects on family satisfaction [55, 61]. High work-home segmentation preferences also alleviated the impact of workplace ostracism on WFC [60]. These findings underscore the critical role of social and personal resources in moderating the relationship between workplace bullying and family satisfaction.

**Marital behaviors.**   Although only a few studies reported marital behaviors and family emotional exhaustion while exploring workplace bullying, the results underscore the multifaceted influence of workplace bullying on family dynamics. For instance, abusive supervision was identified as a social stressor that indirectly triggers anger, withdrawal, and reduced supportive marital behaviors [44]. However, home conflicts reported by employees and their spouses were not directly linked to workplace bullying with affective distress and psychological detachment acting as mediating factors [62].

The underlying mechanisms are grounded in the COR theory, which posits that workplace bullying exhausts an individual's emotional resources, resulting in affective distress that harms spousal interactions and reduces relationship satisfaction [62]. However, psychological detachment is a pivotal resource complementary mechanism with the capacity to alleviate the conflicts at home resulting from workplace bullying [62]. Furthermore, abusive supervision increases social-based WFC by depleting employees' regulatory resources, causing burnout symptoms, and hindering their ability to manage home demands. Subsequently, employees are likely to experience anger, withdrawal, and reduced supportive behaviors toward spouses [44]. These findings highlight the detrimental effects of bullying on marital relationships and the significant role of affective and resource mechanisms in the work-family interface.

**Family emotional exhaustion.**   Only one study in this review explored the impact of workplace ostracism on family emotional exhaustion among employees and their spouses. Applying the work-home resources (W-HR) model, workplace ostracism affects family emotional exhaustion by diminishing positive mood and increasing psychological distress. Both employees' and spouses' family emotional exhaustion primarily arose through psychological distress rather than positive mood. Specifically, workplace ostracism increases employees' psychological distress, which transfers negative emotions and behaviors to their spouses through family undermining, ultimately leading to spouses' family emotional exhaustion [51].

We found two studies that reported how family functioning influences workplace bullying, based on the underlying roles of job commitment and depressive symptoms. Drawing upon COR theory, conflicts in romantic relationships can deplete an individual's emotional resources, leading to depressive symptoms. These symptoms may weaken self-control and self-regulation, increasing the likelihood of aggressive reactions and potentially resulting in abusive supervision [56]. WFC also depletes individual resources, reducing project managers' commitment. This resource depletion and project stress may motivate project managers to engage in abusive supervision [64].

Overall, all the included studies depicted that workplace bullying is directly or indirectly related to indicators of family functioning, and this relationship was consistent across cross-

sectional, experiencing sampling studies and different occupational settings. These findings have important practical and theoretical implications.

## Implications of the findings

### Practical implications

This systematic review has pertinent practical and theoretical implications. As for the practical implications, while most of the results depict consistent relationships between workplace bullying and indicators of family functioning, some slight discrepancies in the findings have to be acknowledged. For instance, customer mistreatment did not directly impact family satisfaction [55], but this family functioning indicator was negatively associated with workplace ostracism and negative workplace gossip [60, 61]. Moreover, bullying was positively linked to self-reported and spousal-reported home conflicts but not directly correlated with relationship satisfaction [62]. These findings emphasize that future research should further examine the relationship between the multiple sources of workplace bullying and various indicators of family functioning for a better understanding.

Regarding the direction of the relationship between bullying and family functioning, 91.2% (n = 34) of the studies investigated the intrusion of workplace dynamics into the family sphere. Researchers examined workplace bullying as a potential predictor of family functioning, and the results consistently affirmed that workplace bullying could trigger employees' WFC, family undermining, erode family satisfaction, as well as increase family emotional exhaustion, anger, and withdrawal behaviors in marital relationships. Meanwhile, only three (8.11%) studies investigated whether this phenomenon contributed to spillover from the home to the workplace. The results confirmed that family functioning could affect abusive supervision both directly and indirectly. That is, family functioning is predictive of abusive supervision. However, it is currently unknown whether and to what extent family functioning can predict an individual's exposure to workplace bullying or ostracism, requiring further exploration.

The present study identified four mechanisms linking workplace bullying to family functioning, comprising emotions, energy (resources), mental health, and work-family relationships. The most extensively studied mediators are negative affect (emotions), WFC, and burnout. In addition, this review found four categories of moderators associated with organizational, social, home, and personal resources (demands). These resources (demands) exacerbate or buffer the bidirectional effects of workplace bullying and family functioning. Various organizations and policy workers may consider perceived organizational support and service training as strategies for addressing workplace bullying and alleviating its effects on employees' family functioning. Furthermore, family support and personal resources such as demographics, coping strategies, psychological states, and personal traits are practical approaches that can be integrated into interventions designed to address workplace bullying at organizational and individual levels. Nevertheless, home resources research has suggested that family support may exacerbate the harmful impact of abusive supervision on psychological distress and mitigate the link between psychological distress and WFC. Pre-existing psychological trauma among employers or supervisors may also affect an individual's resources and their capabilities in dealing with adverse workplace experiences. Further investigation is thus required to determine whether family support can alleviate these adverse outcomes under certain conditions or if additional moderating factors are involved. Insight into these issues will enhance our comprehension of the complex interconnections between workplace bullying and family functioning.

## Theoretical implications

Theoretical frameworks are crucial to build a solid foundation in understanding the mechanisms linking research variables. Several theories were reported in the reviewed studies, which led to the identification of four mechanisms: affect (emotion), energy (resources), mental health, and work-family relationships, linking workplace bullying to family functioning. These moderating and mediating factors were grounded in various theories, with the most frequently used being COR theory, spillover theory, crossover theory, and the work-family interface model. Specifically, the COR theory provided a strong network to elucidate how resource-related (i.e., burnout, resilience, and psychological detachment) and work-family factors influence the direction and strength of the relationship between workplace bullying and family functioning indicators.

The spillover and crossover theories, as well as those relating to WFC also explain how family members can be psychologically harmed by poor treatments and adverse behaviors at the workplace, leading to weakened family relationships. Hence, this review has shown how these theories are interlinked and their suitability in improving our understanding of workplace bullying and family functioning. Despite the diversity of these theories, our findings reflect the opportunities to integrate the models, thoroughly explore, and provide diverse perspectives for understanding the complex relationship between workplace bullying and family functioning. Such integration is essential to explore some of the research gaps identified in this review, particularly the role of family support in ameliorating the adverse effects of workplace bullying and how an individual's vulnerability to this problem can be predicted by family functioning.

## Limitations and future directions

This review has several limitations that should be acknowledged. Firstly, the literature search was confined to the Web of Science, Scopus, PsycINFO, and PubMed databases. While these databases are widely recognized and host vast amounts of scholarly literature, some crucial articles may not be indexed, thus affecting the comprehensiveness of the study findings. Secondly, non-English and gray literature are excluded during the screening process, which may limit the comprehensiveness of the analysis. Consequently, future research should consider expanding the database selection to include high-quality non-English and gray literature while considering constraints such as time and resources to provide a more thorough analysis.

Analysis of the 37 included articles revealed that most studies employed a cross-sectional design, making it difficult to determine causal relationships between bullying and family functioning. Most data predominantly came from self-reported, which carries a bias risk. Future studies could benefit from using a longitudinal design and multi-source data collection, combining self-reports with reports from family members or coworkers to minimize methodological bias and enhance result reliability.

Moreover, since workplace bullying is an occupational risk transcending industries and organizations, it is likely to occur in various sectors. Despite the reviewed studies having covered a range of industries, the findings are restricted to working populations, preferably individuals who are currently employed or have worked in an organizational setting. Hence, the findings are not generalizable to the population of students or younger population. It is essential to specifically target several high-risk organizations or sectors, such as lawyers, the police, and multinational enterprises. We also recommend that researchers make meaningful comparisons of sample data across various industries and sectors to derive more inclusive conclusions. The extrapolation of the results to non-English speaking countries should also be performed cautiously since studies from those locations were not included in the review.

Furthermore, most research has been published over the last decade, suggesting that the field is still nascent, with evident research gaps. Future research could delve deeper into various family outcomes associated with workplace bullying, such as family support, performance, parenting activities, and divorce. Apart from spillover effects, further exploration of crossover effects could widen the social dimensions of parents, children, friends, and coworkers. In terms of the underlying mechanisms, future research could explore cognition mechanisms such as work-related rumination and resource mechanisms, including psychological capital, optimism, and vigor. Furthermore, it would be helpful to examine the extent to which self-efficacy, workplace friendships, and job characteristics alleviate the impact of workplace bullying on family functioning. Examining these variables could provide valuable strategies and insights for mitigating the adverse consequences of workplace bullying on individuals and their families.

Finally, current research has largely evaluated the unidirectional influence of workplace bullying on family functioning. Nevertheless, due to the interconnectedness and permeability of both the work and family domains, an in-depth understanding of how family life influences the workplace experiences of the employee would be valuable. Subsequent research might consider employing the work-home resources (W-HR) model [79] to investigate two-way interactions between the workplace and the family. The work-home resources (W-HR) model has significant strength in emphasizing the bidirectional interactions between the work and family domains.

## Conclusion

This systematic review revealed that the adverse effects of workplace bullying can spill over and impact target family life. Meanwhile, family functioning also influences employees' behaviors at work. The study identified the theoretical foundations, mediators, and moderators of the relationship between workplace bullying and family functioning. Workplace bullying was associated with the family functioning across industries and study design. The longitudinal methodologies are recommended to identify causal links and variable dynamics between workplace bullying and family functioning. Moreover, this relationship requires more in-depth and extensive research regarding its direction, mechanisms, and boundary conditions. Finally, alternative theories that support the links, identifying industries at high risk of bullying, and a model of the family consequences of workplace bullying incorporating existing research results may be potential future research directions.

## Supporting information

**S1 Checklist. PRISMA 2020 checklist.**
(DOCX)

**S1 Table. Search terms and strategy.**
(DOCX)

**S2 Table. Revised checklist for methodological quality.**
(DOCX)

**S1 File. Quality assessment by two authors.**
(ZIP)

**S2 File. Search results from four databases.**
(ZIP)

## Author Contributions

**Conceptualization:** Yang Jie, Daniella Mokhtar.

**Data curation:** Yang Jie, Daniella Mokhtar.

**Formal analysis:** Yang Jie, Daniella Mokhtar.

**Methodology:** Yang Jie, Daniella Mokhtar.

**Project administration:** Daniella Mokhtar.

**Software:** Yang Jie.

**Supervision:** Daniella Mokhtar, Nurul-Azza Abdullah.

**Validation:** Yang Jie.

**Writing – original draft:** Yang Jie, Daniella Mokhtar.

**Writing – review & editing:** Nurul-Azza Abdullah.

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
