## [Decision Letter · Decision Letter 0]

10 Jul 2024

PONE-D-24-22601The relationship between workplace bullying and family functioning: A systematic review.PLOS ONE

Dear Dr. Mokhtar,

Thank you for submitting your manuscript to PLOS ONE. After careful consideration, we feel that it has merit but does not fully meet PLOS ONE’s publication criteria as it currently stands. Therefore, we invite you to submit a revised version of the manuscript that addresses the points raised during the review process.

We look forward to receiving your revised manuscript.

Kind regards,

Rogis Baker, Ph.D

Academic Editor

PLOS ONE

Journal Requirements:

3. Please remove your figures from within your manuscript file, leaving only the individual TIFF/EPS image files, uploaded separately. These will be automatically included in the reviewers’ PDF.

Reviewers' comments:

Reviewer's Responses to Questions

**Comments to the Author**

1. Is the manuscript technically sound, and do the data support the conclusions?

Reviewer #1: Yes

Reviewer #2: Yes

Reviewer #3: Yes

2. Has the statistical analysis been performed appropriately and rigorously? 

Reviewer #1: Yes

Reviewer #2: N/A

Reviewer #3: N/A

3. Have the authors made all data underlying the findings in their manuscript fully available?

Reviewer #1: Yes

Reviewer #2: Yes

Reviewer #3: Yes

4. Is the manuscript presented in an intelligible fashion and written in standard English?

Reviewer #1: Yes

Reviewer #2: Yes

Reviewer #3: No

5. Review Comments to the Author

Reviewer #1: Title: The relationship between workplace bullying and family functioning: A systematic review

Dear Authors,

I enjoyed reading and evaluating your article. You have demonstrated considerable skill and diligence in composing this insightful article. I concur with the authors that workplace bullying is an important topic and its spillover effect on family life should be given more attention, especially the occupational health aspect of workplace bullying should not be overlooked. Your article exhibits notable merits, yet there are some aspects that could be improved. I will present my feedback following the structure of your article.

Summary

The study is a systematic review. The study uses PRISMA framework for analyzing articles from SCOPUS, WOS, PsycINFO, and PubMed. All included studies indicate that workplace bullying is associated with family functioning, both directly and indirectly. The theories which were mostly used for explaining the said relationship were COR, spillover theory, crossover theory, and work-life interface model. Most of the studies included in this systematic review concentrated on one way impact of workplace bullying on family functioning, and the designs used were predominately cross-sectional, non-randomized self-report surveys. The suggested future direction for research were also given with an increased focus on longitudinal designs to better understand the direction, nature, and context of the relationship between workplace bullying and family functioning across different cultures. The authors stressed the need for more extensive research to explore the relationship further.

Introduction

Although the introduction provides a comprehensive overview of the research problem and provide statistics from the global perspective on workplace bullying but some of the information is repeated and the introduction can be made concise. I believe authors can provide more specific examples of existing research gaps by citing more work, this would strengthen the importance of the research. I would suggest research questions and hypotheses can be made clearer and more visible. And lastly, authors can put more emphasis on significance and potential contribution of the study.

PRISMA

The study has largely followed the PRISMA diagram, however some important observations are as follows.

1. Provide more detail on inclusion and exclusion criteria so that reader can quickly understand the selection process (focus on exclusion criteria).

2. Indicate the number of duplicates removed and prominently display in the PRISMA diagram.

3. Distinguish the studies excluded during the title/abstract screening and full text screening. This clarification can clarify the review process.

4. Provide a detail list of reasons for exclusion (I don’t mind if you put it in the Appendix). And briefly discuss this in the main body so the reader can understand the selection process.

5. Were other methods such as citation tracking and expert suggestion/consultation used as sources? If not please mention it briefly in the text.

Search Criteria

The database search method used in this paper seems thorough and methodological, however a few points can be clarified further.

1. Why additional databases such as Google Scholar (very popular these days) were not included in the searching process.

2. What was the logical reason for not including the gray literature?

3. The study mentions the usage of combination of words but does not specify the usage of Boolean operators such as (AND, OR, NOT) in their search strategy. Please provide a documentation for these operators.

4. The inclusion criteria were that studies in English language were include. Why other studies in regional languages were not used? As we may miss out on a big body of literature by excluding those studies. Does workplace bullying only happens in English speaking countries or where English is understood? Please provide a logical reasoning for such a decision.

Discussion

The discussion could benefit from more integration with the broader literature on workplace stress and family dynamics. While the study focuses on workplace bullying, connecting findings to related research on general workplace stress and its family impacts would provide a richer context.

The authors should discuss the methodological limitation as well.

The generalizability issue should be addressed more prominently.

The discussion could benefit from a more detailed exploration of the practical implications of the findings. This could include specific recommendations for organizations, policymakers, and practitioners on how to mitigate the negative impacts of workplace bullying on family functioning.

From where is am looking at the discussion section it seems dense and somewhat crowded. Perhaps breaking it down into subsection can enhance the readability and comprehension (summary of the findings, methodological limitation, theoretical implication, practical implication etc.)

Over all I enjoyed reading this well written paper. Best of Luck

Reviewer #2: This is a timely as well as thoroughly executed systematic review on a topic gaining attention and importance, aiding in the unlocking of knowledge on the ‘anatomy’ and ‘physiology’ of workplace bullying and its effects on families and family life. The complexity of this topic could be characterized as a crosspollination of moderators and dynamics, resulting in a multitude of non-linear ‘processes’.

The authors very well succeeded, in the opinion of this reviewer, in following the chosen route for the review, analyzing and discussing the data from the included papers, resulting in a comprehensive, although readable report.

Additional to the current discussion section, this reviewer suggests, also to widen the scope of the context of the current work. For example the authors could point out the possible effects of pre-existing psychological trauma of workers or employers/supervisors, which can jeopardize these individuals’ resources, e.g., in dealing with adverse workplace experiences.

Also, although this review indicates several work domains, differences between professional domains, e.g., the domain of medicine or law, in which professional identity (professional ‘ego’) might be relevant to point out as well.

(PS - re-reading the manuscript could help solve some last very minor punctuation issues).

Well done and thanks.

WK

Reviewer #3: I enjoyed reading this interesting systematic literature review paper.

The process of identifying papers for reveiw seems systematic and follows PRISMA framework. The writing is clear and flowing well. The structure and headings of the paper seem appropriate. However, my concern is that the theoretical contribution of the paper does not seem to be clear and strong. The authors seem to have described how many studies supported specific relationships between variables of interest. The authors should focus more on the underlying theory being tested and studeid in each paper, and provide theoretical implications emanating from the revie of these studies.

6. PLOS authors have the option to publish the peer review history of their article (what does this mean?). If published, this will include your full peer review and any attached files.

Reviewer #1: **Yes: **Adnan Fateh

Reviewer #2: **Yes: **Wouter Keijser MD PhD

Reviewer #3: No

---

## [Author Response · Author response to Decision Letter 0]

18 Aug 2024

Respected editors and reviwers, I have carefully reviewed and addressed all the comments and suggestions provided by the reviewers and have made the necessary revisions to the manuscript in accordance with the journal’s requirements. A detailed response to each comment has been provided in the “Response to Reviewers” document, which has been uploaded to the system.

I sincerely thank the reviewers and the editorial team for their valuable feedback and constructive suggestions. Your insights have greatly contributed to improving the quality of the manuscript, and I am grateful for your time and effort.

---

## [Editor Report · Decision Letter 1]

29 Aug 2024

The relationship between workplace bullying and family functioning: A systematic review.

PONE-D-24-22601R1

Dear Daniella Mokhtar,

We’re pleased to inform you that your manuscript has been judged scientifically suitable for publication and will be formally accepted for publication once it meets all outstanding technical requirements.

Kind regards,

Rogis Baker, Ph.D

Academic Editor

PLOS ONE
---

## [Editor Report · Acceptance letter]

6 Sep 2024

PONE-D-24-22601R1 

PLOS ONE

Dear Dr. Mokhtar, 

I'm pleased to inform you that your manuscript has been deemed suitable for publication in PLOS ONE. Congratulations! Your manuscript is now being handed over to our production team.

Kind regards, 

on behalf of

Dr. Rogis Baker 

Academic Editor

PLOS ONE